# Mitigating Spurious Correlations
# via Disagreement Probability

**Hyeonggeun Han**[1,2]   **Sehwan Kim**[1]   **Hyungjun Joo**[1,2]
**Sangwoo Hong**[1,2]   **Jungwoo Lee**[1,2,3]*
[1]ECE & [2]NextQuantum, Seoul National University   [3]Hodoo AI Labs
{hygnhan, sehwankim, joohj911, tkddn0606, junglee}@snu.ac.kr

## Abstract

Models trained with empirical risk minimization (ERM) are prone to be biased towards spurious correlations between target labels and bias attributes, which leads to poor performance on data groups lacking spurious correlations. It is particularly challenging to address this problem when access to bias labels is not permitted. To mitigate the effect of spurious correlations without bias labels, we first introduce a novel training objective designed to robustly enhance model performance across all data samples, irrespective of the presence of spurious correlations. From this objective, we then derive a debiasing method, Disagreement Probability based Resampling for debiasing (DPR), which does not require bias labels. DPR leverages the disagreement between the target label and the prediction of a biased model to identify bias-conflicting samples—those without spurious correlations—and up-samples them according to the disagreement probability. Empirical evaluations on multiple benchmarks demonstrate that DPR achieves state-of-the-art performance over existing baselines that do not use bias labels. Furthermore, we provide a theoretical analysis that details how DPR reduces dependency on spurious correlations.

## 1   Introduction

In the realm of machine learning, many classification models employ Empirical Risk Minimization (ERM) [46], which optimizes average performance. However, this approach has been found to underperform on certain groups of data [4, 15, 11] due to the prevalence of spurious correlations within training datasets [31, 38, 39]. Spurious correlations refer to the strong correlations between target labels and easy-to-learn attributes (*i.e.*, bias attributes), which are present in a majority of the training examples. ERM-trained models often rely on these bias attributes [12], leading to biased predictions and poor generalization on minority groups where spurious correlations are absent. For example, consider the cow/camel classification task illustrated in Figure 1. A majority of camel images feature desert backgrounds, while a majority of cow images feature pasture backgrounds. Models trained via ERM might learn to recognize animals based on their backgrounds—desert for camels and pasture for cows—rather than on their distinctive features. This reliance can result in misclassifications, such as erroneously identifying a camel in a pasture as a cow. Addressing these spurious correlations is a

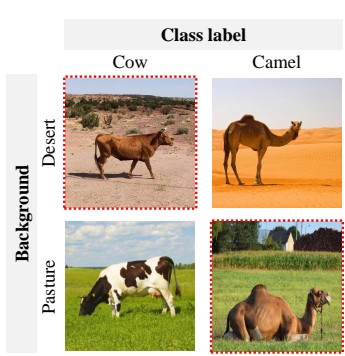

Figure 1: An illustration of the cow/camel classification task. Red dotted boxes indicate samples where spurious correlations do not hold.

*Corresponding author

38th Conference on Neural Information Processing Systems (NeurIPS 2024).

critical issue across various applications, including medical imaging [36], algorithmic fairness [10], and education [37].

There have been extensive efforts to reduce the effects of spurious correlations. Numerous studies have initially relied on the assumption that bias attributes are given in the training dataset [38, 18, 24, 35]. However, annotating bias attributes is labor-intensive and expensive, rendering methods dependent on such annotations impractical. Consequently, research has shifted towards developing debiasing techniques that do not require bias labels during training [34, 27, 1, 29, 28, 49]. In this line of research, samples with spurious correlations are called *bias-aligned samples*, whereas those lacking such correlations are termed *bias-conflicting samples* [34, 27, 23]. In the absence of explicit bias labels, many existing methods focus on identifying bias-conflicting samples and employ strategies such as upweighting or upsampling these samples to counteract the negative impacts of spurious correlations [34, 28, 1]. Despite their straightforward nature, questions persist regarding the optimal scale for the weight of each sample to effectively reduce dependence on spurious correlations.

We thus formulate a debiasing objective for improved robustness against spurious correlations and derive a practical resampling-based algorithm. Specifically, our debiasing objective is built on two groups: the bias-aligned group and the bias-conflicting group, each consisting of bias-aligned and bias-conflicting samples, respectively. This objective encourages the model to perform equally well across both groups. The key insight behind the objective is that models relying on spurious correlations exhibit worse performance on the bias-conflicting group compared to the bias-aligned group, whereas models that do not rely on spurious correlations should exhibit strong performance across both groups. Then, under a simple condition, we derive an objective function as a weighted loss, with weights proportional to the bias-conflicting probability for each training example. Since we consider cases where bias labels are not provided, we use the disagreement probability between the target label and the prediction of an intentionally biased model as a substitute for the bias-conflicting probability. Our proposed approach is simple yet effective in mitigating spurious correlations.

The main contributions of this paper can be summarized as follows:

- We present a debiasing objective for mitigating reliance on bias attributes. This objective aims to guide the model to have similar performance over two groups: the bias-aligned group and the bias-conflicting group. This approach differs from previous works, which typically define groups based on combinations of target labels and bias labels.

- We propose a new method, coined *Disagreement Probability based Resampling for debiasing (DPR)*, which is derived from the proposed objective. DPR leverages the disagreement probability between the target label and the prediction of a biased model to determine the weight of each training example.

- DPR achieves state-of-the-art performance across six benchmarks with spurious correlations, surpassing existing methods that do not use bias labels. Notably, on the bias-conflicting test set of Biased FFHQ (BFFHQ)—which contains only 0.5% of bias-conflicting samples in the training set—the proposed method significantly improves accuracy by 20.87% compared to ERM and by 6.2% compared to the best baseline.

- We theoretically demonstrate that DPR encourages consistent model performance across both bias-aligned and bias-conflicting groups, thus promoting strong performance across all samples regardless of spurious correlations.

## 2 Related work

**Debiasing with bias annotations.** Numerous previous works utilize bias labels for debiasing [38, 48, 18, 3, 44, 53]. For example, Group DRO [38] employs bias labels to define groups and directly enhances worst-group accuracy to mitigate the effect of spurious correlations; meanwhile, LISA [48] mixes two samples with the same label but different domains, or with different labels but the same domain, canceling out spurious correlations and learning invariant predictors. These methods have demonstrated their effectiveness across multiple benchmarks with spurious correlations. However, annotating bias labels for each training example is labor-intensive, and obtaining such labels can sometimes be challenging due to privacy concerns as well [51]. Consequently, in more recent studies, some researchers have utilized only a small set of bias-annotated data to reduce reliance on

bias labels [24, 35, 19]. For example, DFR [24] uses a small group-balanced dataset with bias labels to retrain the last layer of the ERM-trained model.

**Debiasing without bias annotations.** In an effort to eliminate reliance on bias annotations, recent studies predominantly focus on reducing bias without explicit bias labels [34, 22, 27, 1, 29, 28, 49, 45]. Given the unavailability of bias labels, these methods commonly employ a two-stage strategy: (1) identifying bias-aligned and bias-conflicting samples, and (2) training the debiased model by leveraging information obtained from (1). The identification of bias-conflicting samples is accomplished through the use of a deliberately biased model, trained either with generalized cross-entropy loss [34, 22, 27, 1, 29] or standard cross-entropy loss [28, 49, 45]. For instance, LfF [34] and DFA [27] regard samples as bias-conflicting samples if the biased model exhibits higher cross-entropy losses on these samples compared to the debiased model. Subsequent efforts then focus on enhancing classification accuracy for these identified bias-conflicting samples. Numerous studies have proposed diverse debiasing methods [14, 52, 8, 23], including those based on contrastive learning [49, 20] and unsupervised clustering [40, 42].

Many debiasing methods that do not require bias labels aim to enhance performance on the bias-conflicting group, yet the optimal extent of this enhancement remains uncertain. We contend that a debiased model should exhibit consistent performance across both bias-aligned and bias-conflicting groups. However, to the best of our knowledge, no existing training objective is explicitly designed for this purpose. Therefore, we introduce a novel training objective tailored for this purpose, from which we develop DPR—a debiasing method that does not rely on bias labels.

## 3 Problem formulation

We consider a multi-class classification problem with a dataset $\mathcal{D} = \{(x_i, y_i)\}_{i=1}^n$, where each $x_i \in \mathcal{X}$ represents an input, and each $y_i \in \mathcal{Y}$ is the corresponding label from $K$ possible classes. These examples are presumed to be sampled from a training distribution $P$. Given a classification model $f_\theta : \mathcal{X} \to \mathbb{R}^K$ that maps an input to $K$ logits, and a convex loss $\ell : \mathcal{X} \times \mathcal{Y} \to \mathbb{R}_{\geq 0}$, ERM aims to find a model that minimizes the expected loss $\mathbb{E}_{(x,y) \sim P}[\ell(f_\theta(x), y)]$. To this end, we typically minimize a surrogate loss $\hat{\mathcal{L}}_{avg}$:

$$\hat{\mathcal{L}}_{avg} = \frac{1}{n} \sum_{(x,y) \in \mathcal{D}} \ell(f_\theta(x), y). \tag{1}$$

The cross-entropy (CE) loss is commonly used for training classification models. It is defined as $\ell_{\text{CE}}(f_\theta(x), y) = -f_\theta(x)[y] + \log \sum_{y'} \exp(f_\theta(x)[y'])$, where $f_\theta(x)[y]$ denotes the logit corresponding to the $y$-th class label.

We assume there is a spurious correlation presence indicator $b \in \mathcal{B} = \{\text{correlated}, \text{uncorrelated}\}$ in the dataset. As this indicator denotes whether spurious correlations are present within each sample, it can also be considered as a bias-aligned or bias-conflicting group indicator. Similar to the Group DRO setting [38, 17], we adopt the latent prior probability change assumption [43]. With a group indicator $b$, we assume

$$P(x, y|b) = Q(x, y|b), \quad P(b) \neq Q(b), \tag{2}$$

where $P$ and $Q$ denote the training and test data distributions, respectively. Under this assumption, $P(x, y)$ and $Q(x, y)$ are represented as mixtures of the conditional distributions $P(x, y|b)$. Our goal is to find models that are robust against spurious correlations. Regardless of the presence of spurious correlations within each data example, a model that does not rely on these correlations should exhibit strong performance across all examples. In other words, the debiased model should perform consistently well on both bias-aligned and bias-conflicting groups. To this end, we propose the following training objective to avoid spurious correlations:

$$\min_\theta \max_{b \in \mathcal{B}} \left\{ \hat{\mathcal{L}}_b := \frac{1}{n_b} \sum_{(x,y,b) \in G_b} \ell(f_\theta(x), y) \right\}, \tag{3}$$

where $G_b$ is a subset of the training data composed of samples drawn from $P(x, y|b)$ and $n_b$ is the size of $G_b$. The proposed objective aims to minimize the maximum average loss over bias-aligned and

bias-conflicting groups, thereby reducing the reliance of classification models on spurious correlations. However, to utilize the above objective, we must know the information about bias attributes. In the next section, we describe a practical method to train the debiased model using Equation (3) without bias labels.

# 4 DPR: Disagreement Probability based Resampling for debiasing

We present DPR, a resampling method derived from the proposed objective, which does not require bias annotations during training. First, we provide a walk-through of how the objective in Equation (3) can be reformulated as a weighted loss minimization problem. Next, we detail a method for calculating the weight of each training example, along with proxies for both bias-aligned and bias-conflicting groups. Finally, we provide a full description of our algorithm.

## 4.1 Problem reformulation

To utilize the objective in Equation (3) without bias annotations, we reformulate this objective as a weighted loss minimization problem. For this purpose, we introduce the following assumption:

**Assumption 1.** Let $b_a \in \mathcal{B}$ and $b_c \in \mathcal{B}$ represent the bias-aligned and bias-conflicting groups, respectively. The neural network, parameterized by $\theta$, satisfies that $\hat{\mathcal{L}}_{b_a} < \hat{\mathcal{L}}_{b_c}$.

In Assumption 1, we assume that the model, parameterized by $\theta$, exhibits a higher average loss on the bias-conflicting group compared to the bias-aligned group in the training dataset. Under this assumption, the maximum average loss over groups in Equation (3) can be expressed as follows:

$$\max_{b \in \mathcal{B}} \hat{\mathcal{L}}_b = \frac{1}{n_{b_c}} \sum_{(x,y,b_c) \in G_{b_c}} \ell(f_\theta(x), y) \tag{4}$$

$$= \frac{1}{n_{b_c}} \sum_{(x,y,b) \in \mathcal{D}} p(b = b_c|x) \ell(f_\theta(x), y) \tag{5}$$

$$= \frac{1}{n} \sum_{(x,y,b) \in \mathcal{D}} \frac{p(b = b_c|x)}{p(b = b_c)} \ell(f_\theta(x), y). \tag{6}$$

When Assumption 1 is satisfied, Equation (4) holds. Given that all samples in $G_{b_c}$ have $p(b = b_c|x)$ equal to 1, whereas all samples in $G_{b_a}$ have $p(b = b_c|x)$ equal to 0, Equation (5) is derived from Equation (4). As $p(b = b_c)$ is equal to $\frac{n_{b_c}}{n}$, Equation (6) is obtained. By denoting $\frac{1}{n} \cdot \frac{p(b=b_c|x)}{p(b=b_c)}$ as $r(x, y, b)$, we obtain a weighted loss minimization as follows:

$$\min_\theta \sum_{(x,y,b) \in \mathcal{D}} r(x, y, b) \ell(f_\theta(x), y). \tag{7}$$

Note that the weight $r(x, y, b)$ can be interpreted as the sampling probability.

## 4.2 Sampling probability with group proxy

In order to train the model using Equation (7), it is necessary to compute the sampling probability $r(x, y, b)$ for each training example. However, directly calculating the probabilities $p(b = b_c|x)$ and $p(b = b_c)$ is unfeasible without bias labels. To overcome this limitation, we first introduce proxies for the bias-aligned and bias-conflicting groups. We then derive substitutes for the probabilities $p(b = b_c|x)$ and $p(b = b_c)$ using a biased model.

**Group proxy.** We focus on the characteristics of the biased model for the group proxies. Following Nam et al. [34], the biased model $f_\phi$ is trained using ERM with the generalized cross-entropy (GCE) loss [50]:

$$\ell_{\text{GCE}}(f_\phi(x), y) = \frac{1 - p_\phi(y|x)^q}{q}, \tag{8}$$

where $p_\phi(y|x)$ represents the probability assigned to the target label $y$ by the neural network after a softmax layer, and $q \in (0, 1]$ is a hyperparameter. The GCE loss amplifies the bias of the model

**Algorithm 1:** Disagreement Probability based Resampling for debiasing (DPR)

---

**Input:** training set $\mathcal{D}$, biased model $f_\phi$, debiased model $f_\theta$, learning rate $\eta$, the total number of iterations $T_b$ and $T_d$, calibration parameter $\tau$, GCE parameter $q$

```
/* Train the biased model                                               */
```
**for** $t = 1$ **to** $T_b$ **do**
    Sample a mini-batch $\{(x, y)\}$ from $\mathcal{D}$
    Update $\phi$ by training on a mini-batch using Equation (8)
**end**

```
/* Compute the sampling probability                                     */
```
Compute $\hat{r}(x, y)$ for all $(x, y) \in \mathcal{D}$ following Equation (11)

```
/* To satisfy Assumption 1                                              */
```
Initialize the debiased model $f_\theta$ with the biased model $f_\phi$

```
/* Train the debiased model                                             */
```
**for** $t = 1$ **to** $T_d$ **do**
    Sample a mini-batch $\{(x, y)\}$ from $\mathcal{D}$ according to $\hat{r}(x, y)$
    Update $\theta$ by training on a mini-batch using cross-entropy loss
**end**

---

by up-weighting the gradient of the cross-entropy loss for samples with high probability $p_\phi(y|x)$, thereby training the model to rely on spurious correlations. Consequently, the biased model tends to predict correctly for bias-aligned samples and incorrectly for bias-conflicting samples [34]. Building on this insight, leveraging the predictions $y_{\text{bias}}$ of the biased model, we employ the agreement between the label and the biased model's prediction (*i.e.*, $y = y_{\text{bias}}$) as a proxy for the bias-aligned group $b_a$, and the disagreement (*i.e.*, $y \neq y_{\text{bias}}$) as a proxy for the bias-conflicting group $b_c$.

**Sampling probability.** We now discuss how to compute the sampling probability $r(x, y, b)$. Using the group proxies mentioned above, we substitute $p(y \neq y_{\text{bias}}|x)$ for $p(b = b_c|x)$. For a given example $(x, y)$, the disagreement probability $p(y \neq y_{\text{bias}}|x)$ can be computed as follows:

$$p(y \neq y_{\text{bias}}|x) = \sum_{y_{\text{bias}}} p(y, y_{\text{bias}}|x) - p(y = y_{\text{bias}}|x) = 1 - p_{\text{bias}}(y|x), \tag{9}$$

where $p_{\text{bias}}(y|x) = \frac{\exp(f_\phi(x)[y]/\tau)}{Z(\phi)}$ is the probability assigned to label $y$ by the biased model, $Z(\phi) = \sum_{y'} \exp(f_\phi(x)[y']/\tau)$ is the partition function, and $\tau$ is a temperature hyperparameter. Note that $p_{\text{bias}}(y|x)$ is used to approximate $p(b = b_a|x)$. It is crucial for the biased model to accurately capture the spurious correlation structure; hence, the probabilities of the biased model should be appropriately calibrated [13, 33, 32]. We also substitute $p(y \neq y_{\text{bias}})$ for $p(b = b_c)$. This probability can be estimated using all the training data, as in prior works [29, 45]:

$$p(y \neq y_{\text{bias}}) \approx \frac{1}{n} \sum_{(x,y,b) \in \mathcal{D}} p(y \neq y_{\text{bias}}|x) = \frac{1}{n} \sum_{(x,y,b) \in \mathcal{D}} (1 - p_{\text{bias}}(y|x)). \tag{10}$$

With Equations (9) and (10), we compute

$$\hat{r}(x, y) = \frac{1}{n} \cdot \frac{p(y \neq y_{\text{bias}}|x)}{p(y \neq y_{\text{bias}})} = \frac{1 - p_{\text{bias}}(y|x)}{\sum_{(x,y,b) \in \mathcal{D}} (1 - p_{\text{bias}}(y|x))}. \tag{11}$$

Instead of $r(x, y, b)$, we use $\hat{r}(x, y)$ as the sampling probability. We are now ready to train the debiased model.

### 4.3 Training algorithm

We outline the entire training process for our method as follows. First, we train the biased model $f_\phi$. Next, we compute the sampling probability $\hat{r}(x, y)$ using the pretrained biased model. Before proceeding to train the debiased model $f_\theta$, it is worth noting that Equation (7) is derived under Assumption 1. Thus, it is essential to fulfill Assumption 1 when training the debiased model using

Equation (7). We leverage the characteristics of the biased model for this purpose. The biased model typically exhibits higher loss on bias-conflicting samples and lower loss on bias-aligned samples, thereby fulfilling Assumption 1. Consequently, we initialize the model $f_\theta$ with the biased model $f_\phi$ and then train $f_\theta$ using training examples sampled with the probability $\hat{r}(x, y)$. We also employ data augmentation to enhance the diversity of bias-conflicting samples. Simply oversampling these samples without enhancing their diversity does not effectively mitigate bias [27]. Therefore, we enhance the diversity of bias-conflicting samples through data augmentation techniques such as random color jitter and random rotation. The complete training procedure is outlined in Algorithm 1.

## 5 Theoretical analysis

In this section, we theoretically demonstrate that DPR minimizes losses for both bias-aligned and bias-conflicting groups while reducing the disparity between their losses. All proofs are deferred to Appendix A. Let $\mathcal{L}_{avg}$ be the expected average loss:

$$\mathcal{L}_{avg} := \mathbb{E}_{(x,y)\sim P}[\ell(f_\theta(x), y)]. \tag{12}$$

Let $\mathcal{L}_b$ be the average loss of group $b$:

$$\mathcal{L}_b := \mathbb{E}_{(x,y)\sim P_b}[\ell(f_\theta(x), y)], \tag{13}$$

where $P_b = P(x, y|b)$ denotes the training distribution conditioned on $b$, for any $b \in \mathcal{B}$. In this setting, we derive the following inequality for the loss gap between the bias-aligned and bias-conflicting groups.

**Theorem 1.** Suppose that the loss function $\ell(f_\theta(x), y)$ is upper-bounded by a constant $C > 0$. Given two distinct groups $b_a \in \mathcal{B}$ and $b_c \in \mathcal{B}$ such that $b_a \neq b_c$, the following inequality holds with probability at least $1 - \delta$, for any $\delta > 0$:

$$|\mathcal{L}_{b_a} - \mathcal{L}_{b_c}| \leq 2 \cdot \max_{b \in \mathcal{B}} \hat{\mathcal{L}}_b + C \cdot \max_{b \in \mathcal{B}} \sqrt{\frac{8 \log(|\mathcal{B}|/\delta)}{n_b}}. \tag{14}$$

Theorem 1 specifies that the upper bound on the disparity between losses for bias-aligned and bias-conflicting groups is determined by the maximum average loss over groups and a term dependent on the size of the smaller group. Additionally, we derive an inequality associated with the expected average loss.

**Theorem 2.** In the same setting as Theorem 1, the expected average loss is bounded above with probability at least $1 - \delta$:

$$\mathcal{L}_{avg} \leq \max_{b \in \mathcal{B}} \hat{\mathcal{L}}_b + C \cdot \sqrt{\frac{2 \log(1/\delta)}{n}}. \tag{15}$$

According to Theorems 1 and 2, our proposed training objective not only closes the loss gap between bias-aligned and bias-conflicting groups but also reduces the expected average loss. However, in Equation (14), there remains a loss gap due to a term inversely related to the square root of the size of the smaller group. Note that DPR is a resampling method derived from the proposed objective when Assumption 1 is fulfilled, and it identifies and upsamples bias-conflicting samples. Thus, it efficiently minimizes both terms of the upper bound described in Equation (14). Given that $\mathcal{L}_{avg}$ is expressed as a weighted sum of $\mathcal{L}_{b_a}$ and $\mathcal{L}_{b_c}$, these theorems indicate that DPR enhances performance across both bias-aligned and bias-conflicting groups while reducing the performance gap between them.

## 6 Experiments

### 6.1 Datasets

We evaluate the debiasing performance of DPR using six benchmark datasets that exhibit spurious correlations. Colored MNIST and Multi-bias MNIST are synthetic datasets designed under the premise that models learn bias attributes first. Conversely, BAR, BFFHQ, CelebA, and CivilComments-WILDS are real-world datasets where inherent biases degrade model performance. We follow the evaluation protocols of previous studies [1, 27, 28] and provide a detailed description of these datasets in Appendix B.1.

**Colored MNIST.** Colored MNIST (C-MNIST) is a synthetic dataset designed for digit classification, comprising ten digits, each spuriously correlated with a specific color. Following the protocols in Ahn et al. [1], we set the ratios of bias-conflicting samples, denoted as $\rho$, at 0.5%, 1%, and 5% for the training set, and 90% for the unbiased test set. We report the accuracy on this unbiased test set.

**Multi-bias MNIST.** Multi-bias MNIST (MB-MNIST) [1] is a synthetic dataset designed to incorporate more complex patterns compared to C-MNIST and biased-MNIST [41]. MB-MNIST comprises eight attributes: digit [26], alphabet [7], fashion object [47], Japanese character [6], digit color, alphabet color, fashion object color, and Japanese character color. The digit shape is the target attribute, while the other seven serve as bias attributes. In MB-MNIST, bias is introduced by aligning the digit with each of the other seven attributes, each with a probability of $1 - \rho$. In our experiments, $\rho$ is set to 10%, 20%, and 30% for the training set and 90% for the unbiased test set, as in Ahn et al. [1]. We report the accuracy on this unbiased test set.

**Biased action recognition.** The biased action recognition (BAR) dataset [34], designed for action classification, comprises six action classes such as climbing and fishing, each spuriously correlated with a specific place. The training set of BAR contains only bias-aligned samples, while the test set consists solely of bias-conflicting samples. We report the accuracy on this bias-conflicting test set.

**Biased FFHQ.** Biased FFHQ (BFFHQ) is a real-world facial dataset, which has age (young or old) as a label and gender (male or female) as a bias attribute. Predominantly, females are young and males are old in this dataset. We use a bias-conflicting ratio of $\rho = 0.5\%$ in the training set and report accuracies on both an unbiased test set with $\rho = 50\%$ [1] and a bias-conflicting test set with $\rho = 100\%$ [27].

**CelebA.** CelebA [30] is a dataset for facial classification. The goal is to classify celebrity hair color as blond or non-blond, which is spuriously correlated with gender. Notably, only a few blond-haired celebrities are male. Following prior studies [49, 38], we report both average and worst-group accuracies on the test set, where groups are defined as combinations of class labels and bias labels.

**CivilComments-WILDS.** CivilComments-WILDS [5, 25] is a text classification dataset aimed at identifying whether online comments are toxic or non-toxic. The label is spuriously correlated with demographic identities such as gender, race, and religion. Following previous works [28, 25, 49], we report both average and worst-group accuracies on the test set, where groups are defined as combinations of class labels and bias labels.

## 6.2 Experimental setup

**Baselines.** We compare our method with six baselines on various benchmarks: ERM, JTT [28], DFA [27], CNC [49], PGD [1], and LC [29]. ERM denotes conventional training without any considerations for debiasing, while the others are debiasing methods that do not require bias annotations during training.

**Implementation details.** For all datasets except CelebA and CivilComments-WILDS, we follow the experimental settings of Ahn et al. [1]. Specifically, for CMNIST and MB-MNIST, we employ two distinct types of simple CNN models, respectively. For BAR and BFFHQ, we utilize a ResNet18 [16] pretrained on ImageNet [9]. In the case of CelebA, we use a pretrained ResNet50, following the experimental settings of Zhang et al. [49]. For CivilComments-WILDS, we deploy a pretrained BERT model and follow the experimental setup detailed in Liu et al. [28]. Moreover, we apply data augmentation techniques—including random resize crop, random color jitter, and random rotation—for all vision datasets except CelebA, as discussed in Section 4.3. Further details on model architectures, hyperparameters, and image processing are provided in Appendix B.2.

## 6.3 Experimental results and analysis

**Classification accuracy.** Table 1 presents the average accuracies on unbiased test sets for C-MNIST and MB-MNIST. DPR consistently outperforms or matches the performance of other baselines across all experiments with varying bias-conflicting ratios. Notably, for the MB-MNIST dataset with a

Table 1: Average accuracies and standard deviations over three trials on two synthetic image datasets, C-MNIST and MB-MNIST, under varying ratios (%) of bias-conflicting samples. Except for LC, the results of baselines reported in Ahn et al. [1] are provided. The best performances are highlighted in **bold**.

| | C-MNIST | | | MB-MNIST | | |
|---|---|---|---|---|---|---|
| Ratio (%) | 0.5 | 1 | 5 | 10 | 20 | 30 |
| ERM | 60.94 (0.97) | 79.13 (0.73) | 95.12 (0.24) | 25.23 (1.16) | 62.06 (2.45) | 87.61 (1.60) |
| JTT | 85.84 (1.32) | 95.07 (3.42) | 96.56 (1.23) | 25.34 (1.45) | 68.02 (3.23) | 85.44 (3.44) |
| DFA | 94.56 (0.57) | 96.87 (0.64) | 98.35 (0.20) | 25.75 (5.38) | 61.62 (2.60) | 88.36 (2.06) |
| PGD | 96.88 (0.28) | 98.35 (0.12) | **98.62 (0.14)** | 61.38 (4.41) | 89.09 (0.97) | 90.76 (1.84) |
| LC | 97.25 (0.21) | 97.34 (0.16) | 97.44 (0.37) | 25.86 (8.68) | 71.23 (1.71) | 89.57 (2.50) |
| DPR (Ours) | **97.52 (0.33)** | **98.40 (0.03)** | **98.62 (0.12)** | **62.21 (4.02)** | **89.11 (1.65)** | **94.04 (0.26)** |

Table 2: Classification accuracies (%) and standard deviations over three trials on four real-world datasets. Conflicting refers to accuracy on bias-conflicting test sets, while Unbiased indicates accuracy on unbiased test sets. Average and Worst denote average test accuracy and worst-group accuracy, respectively. Results of all baselines except LC are taken from Ahn et al. [1], with the exception of bias-conflicting accuracy on BFFHQ. The best performances are highlighted in **bold**.

| | BAR | BFFHQ | | CelebA | | CivilComments-WILDS | |
|---|---|---|---|---|---|---|---|
| Accuracy (%) | Conflicting | Unbiased | Conflicting | Average | Worst | Average | Worst |
| ERM | 63.15 (1.06) | 77.77 (0.45) | 55.93 (0.64) | 94.9 (0.3) | 47.7 (2.1) | 92.1 (0.4) | 58.6 (1.7) |
| JTT | 63.62 (1.33) | 77.93 (2.16) | 56.13 (0.83) | 88.1 (0.3) | 81.5 (1.7) | 91.1 (-) | 69.3 (-) |
| DFA | 64.70 (2.06) | 82.77 (1.40) | 66.00 (2.00) | - | - | - | - |
| CNC | - | - | - | 89.9 (0.5) | 88.8 (0.9) | 81.7 (0.5) | 68.9 (2.1) |
| PGD | 65.39 (0.47) | 84.20 (1.15) | 70.07 (2.00) | 88.6 (-) | 88.8 (-) | 92.1 (-) | 70.6 (-) |
| LC | 63.45 (2.14) | 83.97 (0.83) | 70.60 (0.60) | - | 88.1 (0.8) | - | 70.3 (1.2) |
| DPR (Ours) | **66.11 (3.29)** | **87.57 (1.22)** | **76.80 (2.51)** | 90.7 (0.6) | **88.9 (0.6)** | 82.9 (0.7) | **70.9 (1.7)** |

Figure 2: Distributions of disagreement probabilities for each sample within bias-aligned and bias-conflicting groups.

ratio of 30%, DPR achieves an unbiased test accuracy of 94.04%, outperforming PGD by 3.28%. Even for MB-MNIST with a ratio of 10%, where all baselines except PGD fail to achieve higher accuracy, DPR exhibits the highest accuracy of 62.21%. In the more complex setting of MB-MNIST, compared to C-MNIST, the effectiveness of DPR is even more pronounced. The superiority of our method is further demonstrated on real-world benchmarks, as shown in Table 2. Our method consistently achieves the best performance for each real-world dataset. Specifically, for the BFFHQ dataset, DPR achieves an accuracy of 87.57% on unbiased test sets, which is 3.37% higher than PGD, and an accuracy of 76.80% on bias-conflicting test sets, which is 6.20% higher than LC. For the CelebA and CivilComments-WILDS datasets, our method achieves the highest worst-group accuracy compared to other baselines, with groups defined as the combination of the target and bias labels. We especially highlight the results for the BFFHQ benchmark, where our method improves accuracy on both unbiased and bias-conflicting test sets. This result supports our claim that DPR enhances performance on both bias-aligned and bias-conflicting groups.

**Identifying group via disagreement probability.** To discern whether each sample belongs to the bias-aligned or bias-conflicting group, we check if the target label disagrees with the prediction of the biased model and use its probability as an up-weight. To evaluate whether the disagreement probability between the target label and the prediction of the biased model distinguishes bias-conflicting samples from bias-aligned samples effectively, we plot the distributions of disagreement probabilities $p(y \neq y_{\text{bias}}|x)$ for each bias-aligned and bias-conflicting sample. The experiment is conducted on the C-MNIST dataset. As illustrated in Figure 2, bias-aligned samples generally exhibit

Table 3: Ablation studies of the proposed method on the C-MNIST and BFFHQ datasets. We report the average test accuracies and standard deviations over three trials on unbiased and bias-conflicting test sets. A checkmark (✓) indicates the case where each component of the proposed method is applied. The best performances are highlighted in **bold**.

| Initialization | GCE | Augmentation | C-MNIST (0.5%) Unbiased | C-MNIST (5%) Unbiased | BFFHQ Unbiased | BFFHQ Conflicting |
|---|---|---|---|---|---|---|
| ✗ | ✗ | ✗ | 66.13 (0.51) | 95.32 (0.42) | 77.07 (1.16) | 54.60 (2.43) |
| ✗ | ✓ | ✓ | 66.47 (1.74) | 95.21 (0.14) | 77.60 (0.79) | 55.60 (1.64) |
| ✓ | ✗ | ✗ | 89.06 (0.62) | 97.36 (0.29) | 78.50 (0.50) | 57.47 (0.90) |
| ✓ | ✓ | ✗ | 95.94 (0.45) | 97.66 (0.17) | 80.93 (1.33) | 63.40 (3.67) |
| ✓ | ✓ | ✓ | **97.52 (0.33)** | **98.62 (0.12)** | **87.57 (1.22)** | **76.80 (2.51)** |

smaller disagreement probabilities compared to bias-conflicting samples. This result demonstrates that disagreement probability effectively differentiates bias-aligned and bias-conflicting samples. Moreover, the relatively high disagreement probability associated with bias-conflicting samples enables DPR to effectively identify and upsample bias-conflicting samples, suggesting that the disagreement probability $p(y \neq y_{\text{bias}}|x)$ serves as an efficient substitute for the bias-conflicting probability $p(b = b_c|x)$.

**Group losses of different models.** As stated earlier in Section 4.3, our debiasing objective in Equation (7) should be used only when Assumption 1 is satisfied. A natural next question is: which type of model satisfies this assumption? We test three types of models: a randomly initialized model, a model pretrained on ImageNet, and a biased model. Specifically, we examine the average loss of a randomly initialized model on the bias-aligned and bias-conflicting groups of C-MNIST with a bias-conflicting ratio of 0.5%. Additionally, we examine the average loss of a pretrained model on the bias-aligned and bias-conflicting groups of BFFHQ. We also investigate the average loss of a biased model on the bias-aligned and bias-conflicting groups of C-MNIST and BFFHQ. The results, shown in Figure 3, indicate that both the randomly initialized and pretrained models have similar average losses on the bias-aligned and bias-conflicting groups. In contrast, the biased model has a higher average loss on the bias-conflicting group compared to the bias-aligned group for both C-MNIST and BFFHQ. This result supports the validity of initializing the debiased model with the biased model.

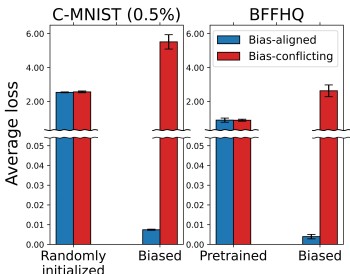

Figure 3: Average loss of randomly initialized, pretrained, and biased models on bias-aligned and bias-conflicting groups. The error bars represent the standard deviations over three trials.

**Ablation study.** We conduct an ablation study on model initialization, GCE, and data augmentation. Model initialization refers to initializing the debiased model with the biased model. The evaluation is performed on C-MNIST with bias-conflicting ratios of 0.5% and 5%, as well as BFFHQ. Table 3 demonstrates the importance of each component. Comparing the first and second rows, we observe that GCE and data augmentation bring small improvements when initialization is not used. However, from rows 3-5, we observe that GCE and augmentation bring significant improvements when initialization is utilized. For BFFHQ, introducing both GCE and augmentation significantly improves the average accuracies on unbiased and bias-conflicting sets by 9.07% and 19.33%, respectively. Furthermore, we observe the performance gap between using and not using initialization. These results suggest that model initialization is crucial and that GCE and augmentation are also important when initialization is introduced. The best performances are consistently achieved when all components are utilized.

## 7 Conclusions and Limitations

We present DPR, a resampling method that leverages disagreement probability to identify and upsample bias-conflicting samples. This method is derived from a novel training objective designed to achieve consistent performance across both bias-aligned and bias-conflicting groups. It does not rely on bias annotations and has demonstrated state-of-the-art performance across spurious correlation benchmarks. However, DPR has certain limitations: its effectiveness depends on how well the biased model captures the spurious correlation structure, as it uses the predictions of this model to infer

the group to which each training sample belongs. Moreover, DPR employs a two-stage approach that complicates the training process and introduces additional hyperparameters. Despite these limitations, DPR's simple implementation and strong performance, supported by theoretical analysis illustrating its ability to reduce loss disparity between groups and minimize average loss, underscore its usefulness in mitigating reliance on spurious correlations.

## Acknowledgements

This work is in part supported by the National Research Foundation of Korea (NRF, RS-2024-00451435(25%), RS-2024-00413957(25%)), Institute of Information & communications Technology Planning & Evaluation (IITP, 2021-0-01059(15%), 2021-0-00106(20%), 2021-0-00180(15%)) grant funded by the Ministry of Science and ICT (MSIT), a grant-in-aid of HANHWA SYSTEMS, Institute of New Media and Communications(INMAC), and the BK21 FOUR program of the Education and Research Program for Future ICT Pioneers, Seoul National University in 2024.

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

# Appendix

## A Missing proofs from Section 5

*Proof of Theorem 1.* Consider the two groups $b_a \in \mathcal{B}$ and $b_c \in \mathcal{B}$ such that $b_a \neq b_c$. Let $d(b_a, b_c)$ be the difference in expected losses between the group $b_a$ and $b_c$:

$$d(b_a, b_c) = |\mathcal{L}_{b_a} - \mathcal{L}_{b_c}|. \tag{16}$$

By the Hoeffding's inequality, the following inequality holds with probability at least $1 - \delta$, for all groups $b \in \mathcal{B}$,

$$\left| \mathcal{L}_b - \hat{\mathcal{L}}_b \right| \leq C \cdot \sqrt{\frac{2 \log (|\mathcal{B}|/\delta)}{n_b}}. \tag{17}$$

Note that the loss $\ell(\cdot)$ is upper-bounded by some constant $C$ according to our assumption and is always non-negative. Accordingly, the following holds with probability at least $1 - \delta$,

$$d(b_a, b_c) \leq \left| \hat{\mathcal{L}}_{b_a} - \hat{\mathcal{L}}_{b_c} \right| + C \left( \sqrt{\frac{2 \log (|\mathcal{B}|/\delta)}{n_{b_a}}} + \sqrt{\frac{2 \log (|\mathcal{B}|/\delta)}{n_{b_c}}} \right). \tag{18}$$

Since the equation $\max\{x, y\} = \frac{x+y}{2} + \frac{|x-y|}{2}$ holds for $x \in \mathbb{R}$ and $y \in \mathbb{R}$, the RHS of Equation (18) is at most:

$$\left| \hat{\mathcal{L}}_{b_a} - \hat{\mathcal{L}}_{b_c} \right| \leq 2 \cdot \max_{b \in \mathcal{B}} \hat{\mathcal{L}}_b. \tag{19}$$

Therefore, we have shown that the following result holds,

$$
\begin{aligned}
d(b_a, b_c) &\leq 2 \cdot \max_{b \in \mathcal{B}} \hat{\mathcal{L}}_b + C \left( \sqrt{\frac{2 \log (|\mathcal{B}|/\delta)}{n_{b_a}}} + \sqrt{\frac{2 \log (|\mathcal{B}|/\delta)}{n_{b_c}}} \right) \\
&\leq 2 \cdot \max_{b \in \mathcal{B}} \hat{\mathcal{L}}_b + C \cdot \max_{b \in \mathcal{B}} \sqrt{\frac{8 \log (|\mathcal{B}|/\delta)}{n_b}}.
\end{aligned}
\tag{20}
$$

This completes the proof of Theorem 1. $\qquad\square$

*Proof of Theorem 2.* By the Hoeffding's inequality, the following inequality holds with at least $1 - \delta$:

$$\left| \mathcal{L}_{\text{avg}} - \hat{\mathcal{L}}_{\text{avg}} \right| \leq C \cdot \sqrt{\frac{2 \log(1/\delta)}{n}}. \tag{21}$$

Accordingly, the expected average loss is bounded with probability $1 - \delta$ as follows:

$$\hat{\mathcal{L}}_{\text{avg}} - C \cdot \sqrt{\frac{2 \log(1/\delta)}{n}} \leq \mathcal{L}_{\text{avg}} \leq \hat{\mathcal{L}}_{\text{avg}} + C \cdot \sqrt{\frac{2 \log(1/\delta)}{n}}. \tag{22}$$

Considering that the training distribution is represented by a mixture of two group distributions $P_{b_a}$ and $P_{b_c}$ for two distinct groups $b_a \in \mathcal{B}$ and $b_c \in \mathcal{B}$, $\hat{\mathcal{L}}_{\text{avg}}$ is equal to

$$k_{b_a} \cdot \hat{\mathcal{L}}_{b_a} + k_{b_c} \cdot \hat{\mathcal{L}}_{b_c} \leq \max_{b \in \mathcal{B}} \hat{\mathcal{L}}_b, \tag{23}$$

where $k_{b_a} = P(b_a)$ and $k_{b_c} = P(b_c)$ denote the prior probability associated with groups. Therefore, $\mathcal{L}_{\text{avg}}$ is upper-bounded with probability at least $1 - \delta$:

$$\mathcal{L}_{\text{avg}} \leq \max_{b \in \mathcal{B}} \hat{\mathcal{L}}_b + C \cdot \sqrt{\frac{2 \log(1/\delta)}{n}}. \tag{24}$$

This completes the proof of Theorem 2. $\qquad\square$

## B  Experimental details

### B.1  Benchmarks

We provide a detailed description of the datasets utilized in Section 6. In Figures 4 to 8, each column corresponds to a distinct class. The images positioned above the dashed line represent bias-aligned samples, whereas those below represent bias-conflicting samples.

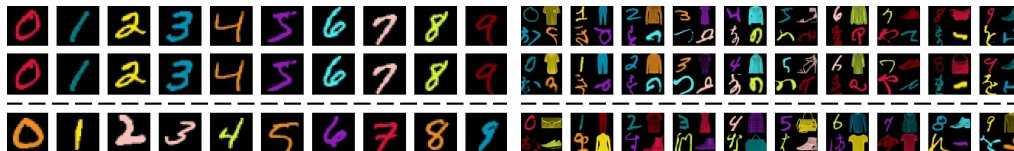

Figure 4: Colored MNIST.        Figure 5: Multi-bias MNIST.

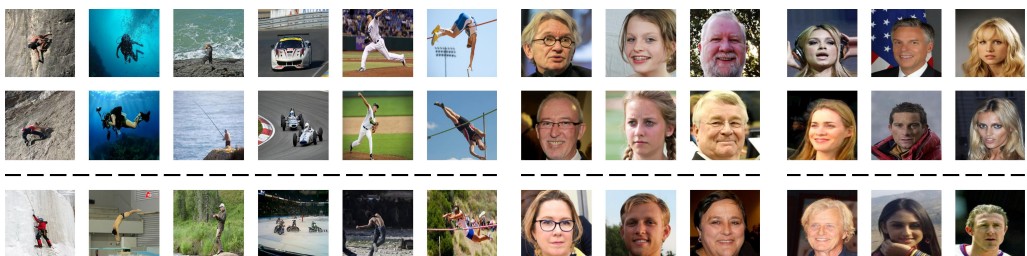

Figure 6: Biased action recognition.   Figure 7: Biased FFHQ.  Figure 8: CelebA.

**Colored MNIST.** Colored MNIST (C-MNIST) is a synthetic dataset designed for digit classification that introduces a controlled spurious correlation between digit shape and color. This dataset consists of ten digits from the MNIST dataset, each associated with a specific RGB color. Following the setting in Ahn et al. [1], ten RGB colors are uniformly selected and applied to the grayscale MNIST images. The image generation involves assigning colors based on the predetermined bias-conflicting ratio, $\rho$. Specifically, each image is colored to create a bias-conflicting sample with probability $\rho$, and a bias-aligned sample otherwise. For bias-aligned samples of a given class $y$, the digit is colored using $c \sim \mathcal{N}(C_y, \sigma I_{3\times3})$, where $C_y \in \mathbb{R}^3$ is the predefined color vector for class $y$. Conversely, bias-conflicting samples receive a color from any class other than $y$, i.e. $C_{U_y} \in \{C_i\}_{i \in [10]\setminus y}$, and colored using $c \sim \mathcal{N}(C_{U_y}, \sigma I_{3\times3})$. The experiments utilize bias-conflicting ratios of 0.5%, 1%, and 5%, with $\sigma$ set to 0.0001. Additionally, we use a 10% of training data as validation data, and an unbiased test set with a bias-conflicting ratio of 90% is employed for performance evaluation.

**Multi-bias MNIST.** Multi-bias MNIST (MB-MNIST) is a synthetic dataset derived from Ahn et al. [1] to incorporate more complex patterns compared to C-MNIST and biased-MNIST [41]. MB-MNIST comprises eight attributes: digit [26], alphabet [7], fashion object [47], Japanese character [6], digit color, alphabet color, fashion object color, and Japanese character color. The digit shape is the target attribute, while the other seven attributes serve as bias attributes. In MB-MNIST, bias is introduced by aligning the digit and its color with probability $1 - \rho$. Similarly, the other six attributes are also independently aligned with the digit, each with probability $1 - \rho$. In our setting, the ratios of bias-conflicting samples are 10%, 20%, and 30% as in Ahn et al. [1]. As CMNIST, we use 10% of the training data as validation data. Test samples are generated with a bias-conflicting ratio of 90% for all bias attributes to create an unbiased test set.

**Biased action recognition.** The biased action recognition (BAR) dataset [34], designed for action classification, is a real-world dataset that encompasses six action classes: climbing, diving, fishing, racing, throwing, and vaulting. Each class is spuriously correlated with a specific place. The training set of BAR consists exclusively of bias-aligned samples, while the test set consists solely of bias-conflicting samples. For example, all the training samples for the climbing class are associated with rock walls, but the test samples for climbing are associated with different settings, such as ice cliffs.

We utilize the data splits defined in Ahn et al. [1], which allocate 10% of the training set to the validation set.

**Biased FFHQ.** Biased FFHQ (BFFHQ) was constructed from the real-world facial dataset FFHQ [21], which has age (young or old) as a label and gender (male or female) as a bias feature. This benchmark was conducted in previous works [27, 1, 22]. Individuals aged between 10 and 29 are labeled as "young", while those aged between 40 and 59 are labeled as "old". In this dataset, the majority of females are young, while males are predominantly old. The training set contains only 0.5% bias-conflicting samples. We report accuracies on both an unbiased test set with a bias-conflicting ratio of 50% [1] and a bias-conflicting test set [27].

**CelebA.** CelebA [30] is a real-world dataset for face classification. Following Sagawa et al. [38], the goal is to classify celebrities' hair color as either blond or non-blond, which exhibits a spurious correlation with gender (male or female). Notably, only a small fraction of blond-haired celebrities are male, which leads to poor performance of ERM-trained models on this group. We employ the training, validation, and test splits as specified in Sagawa et al. [38] and report both the average accuracy and the worst-group accuracy on the test set. In this context, the group is defined as the combination of the class label and the bias label as in prior studies [49, 29, 38].

**CivilComments-WILDS.** CivilComments-WILDS [5, 25] is a dataset for text classification problems. The goal is to classify whether an online comment is toxic or non-toxic, which exhibits a spurious correlation with the mention of certain demographic identities, including male, female, white, black, LGBTQ, Christian, Muslim, and other religions. We use the same data splits as described in Koh et al. [25] and report both the average accuracy and the worst-group accuracy on the test dataset. Here, groups are defined as combinations of class labels and bias labels, as described in previous works [28, 25, 49].

## B.2 Implementation details

We provide descriptions of the implementation details. For CelebA and CivilComments-WILDS, we follow the experimental settings outlined in Zhang et al. [49] and Liu et al. [28], respectively. For the other datasets—CMNIST, MB-MNIST, BAR, and BFFHQ—we follow the experimental setups presented in Ahn et al. [1]. The hyperparameter for GCE, denoted as $q$, is set to 0.7 as in Zhang and Sabuncu [50]. All classification models are trained using an NVIDIA RTX A6000.

### B.2.1 Model architectures and hyperparameters

The model architectures and hyperparameters for each dataset are described:

**C-MNIST.** As in Ahn et al. [1], we utilize a simple CNN architecture (SimConv-1). Please see Appendix B of Ahn et al. [1] for detailed network implementation. We train the model for 100 epochs with SGD optimizer, a batch size of 128, a learning rate of 0.02, weight decay of 0.001, momentum of 0.9, and learning rate decay of 0.1 at every 40 steps. For C-MNIST with bias-conflicting ratios of 0.5% and 1%, we use a temperature of 1; for a ratio of 5%, we use a temperature of 1.1.

**MB-MNIST.** We use a different type of simple CNN model (SimConv-2), following Ahn et al. [1]. Please refer to Appendix B in Ahn et al. [1] for network implementation. We train for 100 epochs with SGD optimizer, a batch size of 32, a learning rate of 0.01, and weight decay of 1e-4, momentum of 0.9. For the MB-MNIST dataset, we use temperatures of 0.9, 1.1, and 1.3 for the ratios of 10%, 20%, and 30%, respectively.

**BAR.** We use a ResNet18 pretrained on ImageNet as in Kim et al. [22]. We train for 100 epochs with SGD optimizer, a batch size of 16, a learning rate of 5e-4, weight decay of 1e-5, momentum of 0.9, learning rate decay of 0.1 at every 20 steps, and a temperature of 0.6.

**BFFHQ.** We utilize an ImageNet-pretrained ResNet18 as in Lee et al. [27]. We train for 160 epochs with Adam optimizer, a batch size of 64, a learning rate of 1e-4, weight decay of 0, learning rate decay of 0.1 at every 32 steps, and a temperature of 0.9.

**CelebA.** We utilize a ImageNet-pretrained ResNet50 and hyperparameters from Zhang et al. [49]. We train for 50 epochs with SGD optimizer, a batch size of 128, a learning rate of 1e-4, weight decay of 0.1, and a temperature of 1.

**CivilComments-WILDS.** We utilize a pretrained BERT with the number of tokens capped at 300 following previous works [49, 28, 1, 25]. We train the biased model using the SGD optimizer without gradient clipping. In contrast, for training the debiased model, we employ the AdamW optimizer with gradient clipping and a linearly-decaying learning rate. Both models are trained for up to 5 epochs with a batch size of 16, a learning rate of 1e-5, weight decay of 0.01, and a temperature of 2.

### B.2.2 Image preprocessing

We employ the same image preprocessing scheme as presented in Ahn et al. [1]. For CMNIST and MB-MNIST, we use fixed sizes of (28×28) and (56×56), respectively. For the remaining datasets, we use a fixed size of (224×224). Additionally, we normalize images from BAR and BFFHQ with a mean of (0.4914, 0.4822, 0.4465) and a standard deviation of (0.2023, 0.1994, 0.2010). For vision datasets except for CelebA, we apply random resize crop, random color jitter, and random rotation to increase the diversity of bias-conflicting samples. To ensure a fair comparison, the same data augmentation techniques are applied to both the baselines and the proposed DPR.

## C Additional ablation study

### C.1 Ablation study on $q$ of GCE

Table 4: Ablation study on $q$ of GCE.

|  | C-MNIST (0.5%) | | MB-MNIST (30%) | |
| --- | --- | --- | --- | --- |
|  | Biased model | Debiased model | Biased model | Debiased model |
| $q = 0.3$ | 26.27 (2.11) | 95.53 (0.74) | 79.73 (0.91) | 95.32 (0.71) |
| $q = 0.5$ | 21.30 (3.10) | 96.83 (0.32) | 83.16 (0.56) | 94.84 (0.15) |
| $q = 0.7$ | 18.55 (2.91) | 97.52 (0.33) | 85.47 (1.26) | 94.04 (0.26) |
| $q = 0.9$ | 16.19 (0.46) | 97.66 (0.06) | 85.26 (0.70) | 93.77 (0.19) |

We conduct an ablation study on the C-MNIST and MB-MNIST datasets, which have bias-conflicting ratios of 0.5% and 30%, respectively, to assess the impact of GCE parameter $q$. As depicted in Table 4, varying $q$ demonstrates distinct effects on the performance of biased and debiased models. For C-MNIST, increasing $q$ enhances the debiased model's performance while degrading that of the biased model. In contrast, for MB-MNIST, decreasing $q$ enhances the performance of the debiased model but worsens the performance of the biased model. These results suggest that the optimal setting of $q$ varies between datasets. Additionally, a consistent observation from both datasets is that as the performance of the biased model decreases, the performance of the debiased model increases. This trend implies that a lower performance of the biased model, indicative of a stronger focus on spurious correlations, leads to more accurate group proxies, which in turn boosts the debiased model's performance.

### C.2 Ablation study on calibration hyperparameter $\tau$

Table 5: Ablation study on calibration hyperparameter $\tau$.

|  | C-MNIST | | MB-MNIST | |
| --- | --- | --- | --- | --- |
| Ratio (%) | 0.5 | 5 | 20 | 30 |
| $\tau = 0.9$ | 97.48 (0.41) | 97.96 (0.28) | 88.10 (1.78) | 91.53 (0.87) |
| $\tau = 1.0$ | **97.52 (0.33)** | 98.38 (0.16) | 88.68 (1.69) | 92.85 (0.65) |
| $\tau = 1.1$ | 96.48 (0.16) | **98.62 (0.12)** | **89.11 (1.65)** | 92.99 (0.66) |
| $\tau = 1.2$ | 95.84 (0.46) | 98.40 (0.18) | 87.80 (2.49) | 93.49 (0.47) |
| $\tau = 1.3$ | 95.07 (0.36) | 98.42 (0.02) | 87.73 (1.53) | **94.04 (0.26)** |

Since DPR utilizes the calibration hyperparameter $\tau$ for capturing the spurious correlation structure well, we conduct an ablation study on C-MNIST and MB-MNIST across various bias-conflicting ratios to assess the effect of $\tau$. Table 5 illustrates how performance varies with different settings of $\tau$. For C-MNIST, the best performances are achieved at $\tau = 1$ and $\tau = 1.1$ for bias-conflicting ratios of

0.5% and 5%, respectively. For MB-MNIST, the best results are obtained with $\tau = 1.1$ and $\tau = 1.3$ for ratios of 20% and 30%, respectively. The optimal $\tau$ can vary not only between different datasets but also according to the extent of the prevalence of spurious correlations within the datasets. These results suggest that adjusting $\tau$ is effective in capturing the diverse spurious correlation structures.

### C.3 Comparison of resampling and reweighting

Table 6: Comparison of resampling and reweighting.

| | C-MNIST | | |
|---|---|---|---|
| Ratio (%) | 0.5 | 1 | 5 |
| Resampling | 97.52 (0.33) | 98.40 (0.03) | 98.62 (0.12) |
| Reweighting | 95.04 (0.35) | 96.21 (0.45) | 97.84 (0.50) |

While reweighting could effectively solve the weighted loss minimization problem presented in Equation (7), DPR adopts resampling. To justify this choice, we conduct experiments to compare resampling with reweighting. Table 6 shows the performance of each method on the C-MNIST dataset across various bias-conflicting ratios. The results reveal that DPR using the resampling strategy consistently outperforms DPR using the reweighting approach. This superiority of resampling over reweighting has been explored and explained by An et al. [2].

