# OpenReview forum: "Mitigating Spurious Correlations via Disagreement Probability"
_NeurIPS.cc/2024/Conference — NeurIPS 2024 poster_

### Official Review · Reviewer_GDUK · 2024-07-10

**Soundness:** 2
**Presentation:** 3
**Contribution:** 2
**Rating:** 5
**Confidence:** 4

**Summary:**

To address the issue of spurious correlations when bias labels are unavailable, the work proposes a new method to mitigate the spurious correlations by minimizing the maximum average loss over bias-aligned and bias-conflicting groups. Additionally, they introduce the disagreement probability/sampling probability, which weights samples in the training objective, to achieve group-robust learning.

**Strengths:**

1. The organization of this paper is good, and it is written in a clear and understandable manner.
2. This work is comprehensive as it covers both theoretical and experimental analyses.
3. The experiments are extensive, considering a rich variety of datasets and baselines.

**Weaknesses:**

1. The method proposed in this paper, DPR, is not novel. Although the authors introduce several concepts such as sampling probability and group indicator, the essence of DPR can be seen as first using ERM with GCE to predict pseudo spurious attribute labels, and then applying the Group DRO method. Similar approaches (pseudo group labels from ERM + invariant learning) can be found in several methods like JTT [1] and CnC [2]. The essence of the author's emphasis on disagreement probability seems to be the predicted probability of ERM that a sample belongs to the minority group (i.e., bias-conflicting groups). Similarly, the reweighting of samples using the predicted probabilities by ERM can be seen in methods like EIIL [3] (the main difference is EIIL focuses more on the regularization term).

2. The use of data augmentation helps alleviate spurious correlations, while other baseline models JTT do not utilize data augmentation, which may lead to unfair comparisons in experiments.

3. The experimental section lacks unity, as evidenced by inconsistent metrics and experimental datasets. It is recommended to focus on the worst group (such as C-MNIST, MB-MNIST) to illustrate the effectiveness of DPR in mitigating spurious correlations. Additionally, I noticed that the average accuracy of CivilComments-WILDS decreases significantly compared to other baselines, which contradicts the theoretical claim in the article of simultaneously reducing the loss for bias-aligned and bias-conflicting groups. Can the authors provide insights into this?

4. I appreciate the authors' demonstration of section  Identifying group via disagreement probability, which shows the effectiveness of biased models in predicting bias-conflicting groups. This is an important evidence for measuring the effectiveness of DPR. However, showcasing it only on C-MNIST is not sufficient. Can you provide experimental results on other datasets, especially real-world datasets?

If authors could provide reasonable replies, I am willing to further increase the score.

[1] Liu, Evan Z., et al. "Just train twice: Improving group robustness without training group information." International Conference on Machine Learning. PMLR, 2021.

[2] Zhang, Michael, et al. "Correct-n-contrast: A contrastive approach for improving robustness to spurious correlations." arXiv preprint arXiv:2203.01517 (2022).

[3] Creager, Elliot, Jörn-Henrik Jacobsen, and Richard Zemel. "Environment inference for invariant learning." International Conference on Machine Learning. PMLR, 2021.

**Questions:**

see my weaknesses.

**Limitations:**

see my weaknesses.

---

> ### Author Rebuttal · Authors · 2024-08-07
>
> We are very grateful for your constructive comments. We have provided answers to each comment. Please let us know if you need any clarification or have additional questions.
>
> > **Q1**: Originality of DPR.
>
> **A1**: The following differences exist between JTT, CNC, and our proposed DPR. JTT identifies misclassified samples and uses heuristic reweighting to upweight them, which does not guarantee to encourage consistent model performance between bias-aligned and bias-conflicting groups, thus having little impact on mitigating the effect of spurious correlations. CNC is a debiasing method using contrastive learning, which aims to close the gap between worst-group loss and average loss. In contrast, our proposed method DPR aims to mitigate the impact of spurious correlations by reducing the performance gap between bias-aligned and bias-conflicting groups.
>
> There is a difference in group definition between EIIL and DPR. Unlike EIIL, which partitions the dataset by maximizing the IRM objective, DPR distinguishes between bias-aligned and bias-conflicting groups based on the presence of spurious correlations, inspired by previous debiasing works [3, 5, 6]. Additionally, there is a difference in implementing a practical algorithm to train a debiased model without bias labels. EIIL optimizes the invariant learning objective, while DPR uses a weighted loss function directly derived from the proposed training objective with easily achievable assumption to reduce performance differences between groups and mitigate spurious correlations. The table below shows the performance of EIIL and DPR, with experiments following the CNC [1] setup, reporting average accuracies and standard deviations over three trials.
>
> |            | CelebA Average | CelebA Worst | CivilComments-WILDS Average | CivilComments-WILDS Worst |
> |:----------:|:--------------:|:------------:|:---------------------------:|:-------------------------:|
> |    EIIL    |   85.7 (0.1)   |  81.7 (0.8)  |          90.5 (0.2)         |         67.0 (2.4)        |
> | DPR (ours) |   90.7 (0.6)   |  88.9 (0.6)  |          82.9 (0.7)         |         70.9 (1.7)        |
>
> As shown, DPR outperforms EIIL across various data modalities.
>
> > **Q2**: Data augmentation setting on other baselines.
>
> **A2**: The experimental setup we used for our experiments is the same as the experimental settings of CNC [1] for CelebA, JTT [2] for CivilComments-WILDS, and PGD [3] for the remaining datasets, which include data augmentation. Therefore, the same data augmentation technique was applied to all other baselines, including JTT, as well as our proposed algorithm.
>
> > **Q3**: lack of unity in metrics and contradictory results on CivilComments-WILDS.
>
> **A3-1**: Lack of unity in metrics
>
> We appreciate your recommendation. As you suggested, we could focus on evaluating the worst group to show how effective DPR is in mitigating spurious correlations. We believe we can demonstrate this using existing experimental and dataset settings that focus on the worst group.
>
> To demonstrate the effectiveness of DPR, we selected six datasets (two synthetic and four real datasets) widely used in existing debiasing papers. The metrics used in our experiments are those commonly employed in previous debiasing studies [1-4], chosen according to each dataset's specific setting.
>
> For datasets not focused on the worst group, the bias-conflicting ratio in the training dataset is relatively low (such as 0.5%, 5%, or 10%), while the bias-conflicting ratio in the test set used for performance reporting is relatively high (such as 50%, 90%, or 100%). Therefore, we believe this provides a sufficient metric to evaluate how effectively a debiasing method mitigates spurious correlations.
>
> **A3-2**: Contradictory results on CivilComments-WILDS.
>
> The practical algorithm DPR, derived from the training objective with an assumption, initializes the debiased model with a biased model and then trains it by oversampling bias-conflicting samples to obtain the debiased model. If, during the training process of oversampling bias-conflicting samples, the model forgets previously learned information about bias-aligned samples, this could result in an accuracy drop and consequently lower average accuracy. Unlike CelebA, where the average accuracy of DPR is higher than other baselines except ERM, a relatively lower average accuracy is reported in CivilComments-WILDS. This could be due to certain characteristics of the BERT model and language data. Analyzing and addressing these causes will be part of future work.
>
> > **Q4**: Group identification experimental results on real-world datasets.
>
> **A4**: We conducted an experiment on group identification via disagreement probability for the real-world dataset BFFHQ. The figures showing the experimental results have been uploaded as a PDF file in the global response. Please refer to it. As shown in the figures, for BFFHQ, bias-aligned samples generally exhibit smaller disagreement probabilities compared to bias-conflicting samples, effectively distinguishing between bias-aligned and bias-conflicting samples. Additionally, the relatively large disagreement probability of bias-conflicting samples allows DPR to effectively identify and upsample these bias-conflicting samples.
>
> [1] Correct-n-contrast: A contrastive approach for improving robustness to spurious correlations, ICML 2022
>
> [2] Just train twice: Improving group robustness without training group information, ICML 2021
>
> [3] Mitigating dataset bias by using per-sample gradient, ICLR 2023
>
> [4] Learning debiased representation via disentangled feature augmentation, NeurIPS 2021
>
> [5] Learning from failure: De-biasing classifier from biased classifier, NeurIPS 2020
>
> [6] Learning debiased classifier with biased committee, NeurIPS 2022

---

> ### Comment · Reviewer_GDUK · 2024-08-08
>
> Dear Author,
>
> Thank you for your response and the additional experimental results.
>
> Regarding Q1, although EIIL and DPR use different terminologies, in the presence of spurious features, EIIL encourages the model to violate the principles of invariant learning by maximizing the EI objective in the first stage, thereby learning the spurious attribute labels of the samples (i.e., biased model). The model probabilities (in a binary classification scenario) are then used as sample weights for invariant learning in the next stage (i.e., debiased model). Despite differences in algorithm details, EIIL is close to DPR.
>
> For Q2, could you please confirm where JTT and CnC indicate that they use data augmentation techniques? For instance, in the case of JTT, the original paper explicitly states in Section A Training Details that the experimental settings for Waterbirds and CelebA are "no data augmentation" (please correct me if I am wrong).
>
> Best,
>
> Reviewer GDUK

---

> > ### Author Response · Authors · 2024-08-09
> >
> > We appreciate your comments and provide answers to each question.
> >
> > For Q1, the differences between EIIL and the proposed DPR are as follows:
> >
> > First, in terms of the algorithm, there are distinct differences between EIIL and DPR. The most significant difference is that in DPR, the weighted loss function used to train the debiased model is mathematically derived directly from the training objective under an easily achievable assumption. We also utilized the characteristics of the biased model (i.e., relatively low loss for the bias-aligned group) to satisfy this assumption, ensuring that the loss function used to train the debiased model aligns with the training objective. This process of deriving the loss function and the resulting loss function itself are novel and have not been proposed in previous debiasing papers, including EIIL. Therefore, this newly derived loss function leads to a new algorithm, DPR, which is distinct from EIIL. Although it may use similar components (i.e., biased model, reweighting), DPR creatively combines these components, as it is derived by the newly derived loss function.
> >
> > Secondly, the group definitions used by EIIL's objective function and our proposed objective function are different. Upon examining EIIL's objective function and algorithm, it seems that EIIL has no restrictions on the number of groups (i.e., environments) and this is not explicitly specified. In contrast, we explicitly defined two groups - bias-aligned and bias-conflicting - and based our objective function on these two groups. For example, regardless of whether a dataset consists of 6 classes or 10 classes, or irrespective of the dataset's composition and characteristics, we propose dividing it into two groups in every situation and building our objective function upon these two groups. We would like to point out that there is a difference between our approach and EIIL's, as EIIL did not start by dividing into two groups to set up its objective function, but rather proposed an objective function and algorithm for an arbitrary number of groups.
> >
> > The differences mentioned above have indeed led to performance disparities. As you can see in the comparison experiment table above for CelebA and CivilComments-WILDS, there is a significant performance gap, showing that our DPR outperforms EIIL. We believe this highlights the differences from EIIL and demonstrates the superiority of our method.
> >
> > For Q2, we followed their experimental settings, including data augmentation policies (e.g., whether to use it, types, etc.). Therefore, since JTT and CNC did not use data augmentation for CivilComments-WILDS and CelebA, we also refrained from using it.

---

> > > ### Comment · Reviewer_GDUK · 2024-08-10
> > >
> > > Dear author,
> > >
> > > Thank you for your detailed reply.
> > >
> > > After reading the reply and the opinions of other reviewers, I choose to keep my score. I still think the novelty of this work is limited. Besides, I am concerned about the fairness of the experiments. As far as I know, data augmentation (which helps alleviate spurious correlations) was not used for the baselines such as JTT and CnC on CelebA and CivilComments-WILDS. I suggest the author should further explain in detail in the main text or appendix what experimental settings they used to ensure that the comparison is reasonable. To sum up, I choose to maintain my score.
> > >
> > > Best,
> > >
> > > Reviewer GDUK

---

> > > > ### Author Response · Authors · 2024-08-10
> > > >
> > > > Thank you for taking the time to read my response and reply. Additionally, I would like to clarify something regarding the experimental setup. When we said the experimental settings were the same, we meant that comparative experiments were conducted in the same experimental environment as the baselines. Although different data augmentation policies may have been applied to each dataset, all baselines, including our proposed DPR, were tested using the same data augmentation policy for each dataset. In other words, for CelebA and CivilComments-WILDS, no data augmentation was used for all other baselines including JTT and CNC, as well as the proposed DPR. For other datasets including BFFHQ, experiments were conducted using data augmentation for all baselines and the proposed DPR. This setting was used to ensure fairness in comparative experiments. As you suggested, we will include this information in the appendix of the revised version. We appreciate your careful comments.

---

> > > > > ### Comment · Reviewer_GDUK · 2024-08-10
> > > > >
> > > > > Thank you for clarifying the experiment. Please add the description of the experimental setting in the revised version. According to the author's explanation, I am willing to increase my score.

---

> > > > > > ### Author Response · Authors · 2024-08-10
> > > > > >
> > > > > > We sincerely thank you for your insightful suggestions and the decision to raise the score. We will include the detailed experimental settings in the revised version.

---

### Official Review · Reviewer_Het1 · 2024-07-11

**Soundness:** 3
**Presentation:** 3
**Contribution:** 3
**Rating:** 6
**Confidence:** 3

**Summary:**

The authors mainly target fairness without accessing bias labels. They suggest a new learning objective that minimizes the loss of the bias group showing the highest ERM and demonstrate that minimizing this objective decreases the upper bound of the expected average loss. To utilize this loss when the bias labels are not accessible, they derive the oversampling method from the new learning objective. In experiments, they show that their method outperforms baselines on various datasets.

**Strengths:**

* They suggest a new learning objective and provide theoretical support for the proposed objective.
* The authors derive the sampling weights from the perspective of the new learning objective.
* They demonstrate that their method outperforms baselines on various benchmark datasets.

**Weaknesses:**

* Since the sampling weights are derived from the proposed learning objective, it's important to demonstrate the effectiveness of the proposed learning objective itself. Could you confirm if it is superior to GroupDRO in scenarios where bias labels are provided on CelebA?
* The effectiveness of the proposed method compared to baselines seems weak. Except for the synthetic datasets and BFFHQ, their performance gain is marginal.
* There is no clear explanation of how to choose hyperparameters or select the model from a specific epoch during training. In the absence of an unbiased validation set, such as BFFHQ, how were these selections made?
* In situations where the ratio of bias-conflicting samples is low, using oversampling methods might make training unstable. Could you show how the test accuracy of bias-conflicting samples changes every 10 epochs during the whole training process on BFFHQ (160 epochs as mentioned in the Appendix)?

**Questions:**

* According to Section 4.3, the authors introduce additional augmentation for the diversity of minor class samples. Did the other algorithms compared in the experiment also use these augmentation techniques?

**Limitations:**

The authors properly address the limitations in Section 7.

---

> ### Author Rebuttal · Authors · 2024-08-07
>
> We thank the reviewer for the constructive comments. We have provided answers to each comment. Please let us know if you need any clarification or have additional questions.
>
> > **Q1**: The effectiveness of the proposed learning objective itself.
>
> **A1**: As you mentioned, to demonstrate the effectiveness of the proposed learning objective, we conducted comparative experiments with GroupDRO in a scenario where bias labels are provided for CelebA. We followed the setting of CNC [1] and reported the average accuracies (%) and standard deviations over three trials. The results are as follows.
>
> |          | CelebA Average | CelebA Worst |
> |:--------:|:--------------:|:------------:|
> | GroupDRO |   93.9 (0.1)   |  88.9 (1.3)  |
> |   Ours   |   92.3 (0.8)   |  90.9 (0.6)  |
>
> As shown in the table above, when bias labels are provided, the proposed learning objective (Ours) demonstrates excellent performance on CelebA. This clearly demonstrates the effectiveness of the objective.
>
> > **Q2**: The effectiveness of the proposed method compared to baselines.
>
> **A2**: The proposed DPR consistently outperforms or matches other baselines across 11 metrics on all six benchmarks. Considering that evaluations were conducted on different settings with various bias-conflicting ratios, types of bias, and data modalities, DPR's consistently highest performance across all benchmarks clearly demonstrates that it is more effective than any of the other baselines.
>
> > **Q3**: Hyperparameter and model selection criteria.
>
> **A3**: We followed the experimental settings of CNC [1] for CelebA, JTT [2] for CivilComments-WILDS, and PGD [3] for the remaining datasets, which include their hyperparameters. In other words, all hyperparameters are the same as those of the aforementioned works [1, 2, 3], except for the temperature hyperparameter newly introduced by our proposed method, DPR.
>
> Following existing debiasing papers [1, 2, 4, 5, 6], we selected the temperature and performed early stopping based on the best worst-group validation accuracy for CelebA and CivilComments, and the best validation accuracy for the remaining datasets.
>
> > **Q4**: The test accuracy on BFFHQ bias-conflicting samples every 10 epochs.
>
> **A4**: As you requested, we conducted an experiment to examine the changes in the test accuracy on bias-conflicting samples of BFFHQ. We reported the average accuracies (%) and standard deviations over three trials. The results are as follows.
>
> |      Epoch     |       1      |      10      |      20      |      30      |      40      |      50      |      60      |      70      |      80      |      90      |      100     |      110     |      120     |      130     |      140     |      150     |      160     |
> |:--------------:|:------------:|:------------:|:------------:|:------------:|:------------:|:------------:|:------------:|:------------:|:------------:|:------------:|:------------:|:------------:|:------------:|:------------:|:------------:|:------------:|:------------:|
> | BFFHQ Conflict | 76.67 (0.31) | 72.40 (3.80) | 74.87 (2.53) | 76.40 (0.92) | 76.60 (4.01) | 75.67 (3.07) | 75.00 (0.80) | 75.87 (1.68) | 75.60 (1.51) | 75.87 (1.36) | 76.07 (1.42) | 75.80 (1.51) | 75.87 (1.42) | 75.87 (1.86) | 75.80 (1.78) | 76.07 (1.62) | 76.07 (1.50) |
>
> As can be seen in the table above, although the test accuracy shows somewhat unstable patterns during the training process, it achieves a test accuracy close to 76% for many epochs. This figure consistently outperforms other baselines.
>
> > **Q5**: Whether the same data augmentation was applied to other baselines.
>
> **A5**: We conducted our experiments using the same experimental setup as Ahn et al. [3], which includes the data augmentation explained in Section 4.3 of the paper. In other words, the same data augmentation technique was applied not only to our proposed DPR but also to all other baselines.
>
> [1] Correct-n-contrast: A contrastive approach for improving robustness to spurious correlations, ICML 2022
>
> [2] Just train twice: Improving group robustness without training group information, ICML 2021
>
> [3] Mitigating dataset bias by using per-sample gradient, ICLR 2023
>
> [4] Distributionally robust neural networks for group shifts: On the importance of regularization for worst-case generalization, ICLR 2020
>
> [5] Learning from failure: De-biasing classifier from biased classifier, NeurIPS 2020
>
> [6] Learning debiased classifier with biased committee, NeurIPS 2022

---

> > ### Comment · Reviewer_Het1 · 2024-08-12
> >
> > Thank you for providing a detailed response. Most of my concerns are resolved except for the hyperparameter selection. In A3, I hope to confirm whether the authors used the best validation accuracy on the BFFHQ, where the validation set is highly biased, for hyperparameter selection. Given that the validation set is biased, relying on its accuracy for hyperparameter tuning might not yield reliable results.

---

> > > ### Author Response · Authors · 2024-08-12
> > >
> > > Thank you for your comments, and we would like to provide a response to your question.
> > >
> > > Yes, as you mentioned, we tuned the hyperparameters and reported the performance using a highly biased validation set. The reasoning behind this approach is as follows:
> > >
> > > 1. about the use of bias-aligned validation set
> > >
> > >     **First, the results of other baselines reported for BFFHQ were taken from Ahn et al. [1]. To ensure a fair comparison, we conducted our experiments on BFFHQ following the experimental settings of Ahn et al. [1]. These settings include the dataset split.** Therefore, we used the same dataset split for our experiments and reported the performance of the proposed DPR using a bias-aligned validation set (composed only of bias-aligned samples).
> > >
> > > 2. about the reliability of the performance of DPR
> > >
> > >     Even though we selected hyperparameters and reported performance using a bias-aligned validation set, DPR outperforms other baselines. We believe that the reason for these results is that **DPR is a debiasing method derived from a newly suggested training objective that aims to improve performance for both bias-aligned and bias-conflicting groups while reducing the performance gap between them.** This explains why DPR outperforms other baselines, even when performance is reported using a bias-aligned validation set. While the performance of DPR could improve further if evaluated using an unbiased validation set or a bias-conflicting validation set with the same distribution as the test set, we used the same dataset split as Ahn et al. [1] for a fair comparison.
> > >
> > >
> > > [1] Mitigating dataset bias by using per-sample gradient, ICLR 2023

---

> > > > ### Comment · Reviewer_Het1 · 2024-08-13
> > > >
> > > > Thank you for the detailed response. All of my concerns are resolved. I raise my score to weak accept.

---

> > > > > ### Author Response · Authors · 2024-08-13
> > > > >
> > > > > Thank you for taking the time to read our responses and provide comments on them. We are pleased that all concerns have been resolved. Additionally, we are extremely grateful for your decision to raise the score.

---

### Official Review · Reviewer_1RoL · 2024-07-12

**Soundness:** 4
**Presentation:** 4
**Contribution:** 4
**Rating:** 8
**Confidence:** 4

**Summary:**

This paper addresses the critical issue of spurious correlations in machine learning models, where models often rely on easy-to-learn bias attributes rather than the intended target features, leading to poor generalization on minority groups where these spurious correlations are absent.

The authors propose a novel method called Disagreement Probability based Resampling for debiasing (DPR), which aims to mitigate the effects of spurious correlations without requiring explicit bias labels during training. The key contributions of this paper are:

1. A new training objective designed to achieve consistent model performance across both bias-aligned and bias-conflicting groups, encouraging robustness against spurious correlations.

2. Development of DPR, a practical resampling method derived from the proposed objective. DPR leverages the disagreement probability between the target label and the prediction of a biased model to identify and upsample bias-conflicting samples.

3. Theoretical analysis demonstrating that DPR minimizes losses for both bias-aligned and bias-conflicting groups while reducing the disparity between their losses.

4. Extensive empirical evaluation on six benchmark datasets (including both synthetic and real-world data) showing that DPR achieves state-of-the-art performance compared to existing methods that do not use bias labels.

5. Ablation studies and analyses that provide insights into the effectiveness of various components of the proposed method, such as model initialization, generalized cross-entropy loss, and data augmentation.

**Strengths:**

### Originality:

1. Novel objective formulation: The authors introduce a new training objective designed to achieve consistent performance across bias-aligned and bias-conflicting groups. This approach differs from previous works that typically define groups based on combinations of target labels and bias labels.
2. Creative use of disagreement probability: The method leverages the disagreement between target labels and biased model predictions as a proxy for identifying bias-conflicting samples. This is an innovative way to address the challenge of not having explicit bias labels.
3. Unique combination of existing ideas: DPR creatively combines ideas from biased model training, resampling techniques, and theoretical analysis to create a cohesive and effective debiasing method.

### Quality:

1. Comprehensive empirical evaluation: The authors test their method on six diverse benchmarks, including both synthetic and real-world datasets, providing a thorough assessment of DPR's performance.
2. Theoretical foundations: The paper includes a rigorous theoretical analysis that supports the empirical results, demonstrating how DPR reduces loss disparity between groups and minimizes average loss.
3. Ablation studies: The authors conduct detailed ablation studies to understand the contribution of each component of their method, showing a commitment to thorough scientific investigation.
4. Comparison with state-of-the-art: DPR is compared against multiple recent baselines, consistently showing superior or comparable performance.

### Clarity:

1. Clear problem formulation: The authors provide a clear definition of the problem and their approach to solving it.
2. Step-by-step derivation: The development of DPR from the initial objective to the final algorithm is presented in a logical, easy-to-follow manner.
3. Visual aids: The paper includes helpful figures and tables that illustrate key concepts and results.
4. Detailed experimental setup: The authors provide comprehensive information about their experimental setup, facilitating reproducibility.

### Significance:

1. Addressing a crucial challenge: Spurious correlations are a major issue in machine learning, and this work provides a novel approach to mitigating them without requiring bias labels, which is often a practical constraint in real-world scenarios.
2. Broad applicability: The method is shown to be effective across various types of data (image and text) and problem setups, suggesting its potential for wide adoption in different domains.
3. Theoretical and practical relevance: By providing both theoretical guarantees and strong empirical results, the paper bridges the gap between theory and practice in addressing spurious correlations.
4. Potential impact on fair ML: The proposed method could contribute to the development of fairer and more robust machine learning models, which is a critical goal in the field.

**Weaknesses:**

Nothing

**Questions:**

Nothing

**Limitations:**

Authors have discussed the limitations sufficiently.

---

> ### Author Rebuttal · Authors · 2024-08-07
>
> Thank you for pointing out the strength and originality of DPR, which uses a loss function directly derived mathematically from setting an objective. We also appreciate your high regard for the various experiments, theoretical analyses, and ablation studies that support the effectiveness of DPR. We would like to express our deep gratitude once again for your positive review.

---

> ### Comment · Reviewer_1RoL · 2024-08-08
> **Thanks for your response**
>
> I want to state that I have also checked other reviewers' concerns and questions and I believe they are well-answered. I will keep my score. Thanks for answering questions and clarifying strengths and limitations.

---

> ### Author Response · Authors · 2024-08-09
>
> Thank you very much for your comments and for taking the time to review our responses. Additionally, We really appreciate your review and your decision to maintain your score.

---

### Official Review · Reviewer_qVjP · 2024-07-15

**Soundness:** 2
**Presentation:** 2
**Contribution:** 2
**Rating:** 5
**Confidence:** 5

**Summary:**

This paper proposes a re-sampling approach based on disagreement between bias predictions and target label predictions. First, a biased model is trained using generalized cross-entropy. Then, sample-wise weights are determined by calculating the probability of disagreement between the bias predictions of the biased model and the target label predictions of the target model.

**Strengths:**

1. This paper tackles the significant and pressing issue of robust learning, which is crucial for effectively using trained models in real-world applications.

2. The proposed sampling method is both simple and straightforward, making it easy to understand and implement.

**Weaknesses:**

1. The method extends the approach of upweighting misclassified samples in the existing Just Train Twice (JTT) by utilizing generalized cross-entropy (GCE) and the disagreement between bias predictions and target label predictions. However, this extension does not appear novel. Furthermore, do there not exist approaches that use bias predictions? If so, discussing these studies would provide context and underscore the novelty of the proposed method.

2. Color jitter is employed as a data augmentation technique that can directly solve the color bias in CMNIST. However, the ablation study does not show significant performance improvement. What is the reason for this outcome? Additionally, for a fair evaluation, the same data augmentation should be applied to the baselines.

3. CMNIST is a relatively easy dataset, and BAR is not widely used recently. Therefore, additional experiments on other widely used datasets, such as CIFAR-10C, Waterbird, or NICO, are needed. Demonstrating similar performance improvements on these datasets would better support the experimental effectiveness of the proposed method.

4. The related works section does not discuss some recent studies, such as [1, 2].

[1] Lee et al., "Surgical fine-tuning improves adaptation to distribution shifts.", ICLR'23

[2] Jung et al., "Fighting Fire with Fire: Contrastive Debiasing without Bias-free Data via Generative Bias-transformation", ICML'23

**Questions:**

Please see the Weaknesses

**Limitations:**

Please see the Weaknesses

---

> ### Author Rebuttal · Authors · 2024-08-07
>
> Thank you for your insightful comments. We've carefully considered them and provided responses. Please let us know if you need any clarification or have additional questions.
>
> > **Q1**: The differences between the proposed DPR and other existing approaches using bias predictions.
>
> **A1**: As outlined in Section 2, several debiasing methods utilize biased models to identify bias-conflicting samples, including JTT [1], DFA [2], PGD [3], and LC [4]. Each method employs a unique approach:
>
> 1. JTT upweights bias-conflicting samples by specifying their weights as a hyperparameter.
>
> 2. DFA generates diverse bias-conflicting samples by mixing features between samples, using these to mitigate spurious correlations.
>
> 3. PGD, a resampling-based method, determines upweighting based on the sample gradient of a biased model.
>
> 4. LC mitigates spurious correlations by correcting the sample logit.
>
> Our proposed DPR method differs from these existing approaches in several key aspects:
>
> 1. DPR is derived from a newly suggested learning objective (equation 3) that aims to improve performance for both bias-aligned and bias-conflicting groups while reducing the performance gap between them.
>
> 2. In contrast, JTT's heuristic reweighting strategy may not consistently achieve this objective, potentially limiting its effectiveness in mitigating spurious correlations.
>
> 3. PGD interprets debiasing as a min-max problem of minimizing loss for maximally difficult samples, which can be relaxed to minimizing the trace of inverse Fisher Information.
>
> 4. LC focuses on debiasing by maximizing group-balanced accuracy.
>
> These differing objectives distinguish our approach from existing methods. In section 6.3, we provide empirical evidence demonstrating that our proposed method outperforms other baselines including JTT, DFA, PGD, and LC across all six benchmarks.
>
> > **Q2**: Data augmentation setting and the impact of color jitter on mitigating color bias.
>
> **A2**: We would like to clarify that our experiments followed the same setup as Ahn et al. [3], including the data augmentation settings. This ensures a fair comparison, as the same data augmentation techniques were applied to all baseline models as well as our proposed method.
>
> Regarding the C-MNIST dataset results presented in Table 3 of our paper:
>
> 1. Our model achieves high performance (95.94% and 97.66%) on C-MNIST with just initialization and the use of GCE loss, even without augmentation. This is due to the effectiveness of the biased model trained with GCE loss in identifying and oversampling bias-conflicting samples within the C-MNIST dataset, thus reducing the impact of color bias.
>
> 2. As you correctly pointed out, incorporating data augmentation further addresses the color bias in C-MNIST. This additional step effectively mitigates the remaining bias, leading to significant performance improvements.
>
> > **Q3**: Additional experiments on other widely used datasets.
>
> **A3**: We conducted additional experiments on CIFAR-10C following the experimental settings of Ahn et al. [3] and reported the average accuracies (%) and standard deviations over three trials. The results are as follows.
>
> |            | CIFAR-10C (0.5%) | CIFAR-10C (1%) | CIFAR-10C (5%) |
> |:----------:|:----------------:|:--------------:|:--------------:|
> |     ERM    |   23.06 (1.25)   |  25.94 (0.54)  |  39.31 (0.66)  |
> |     JTT    |   25.34 (1.00)   |  33.62 (1.05)  |  45.13 (3.11)  |
> |     DFA    |   29.96 (0.71)   |  36.35 (1.69)  |  51.19 (1.38)  |
> |     PGD    |   30.15 (1.22)   |  42.02 (0.73)  |  52.43 (0.14)  |
> | DPR (Ours) |   32.20 (0.81)   |  43.77 (0.93)  |  53.10 (0.62)  |
>
> The ratios shown in parentheses next to the dataset names in the table represent the bias-conflicting ratio in the training set. As shown in the table above, the proposed DPR consistently outperforms other baselines on CIFAR-10C with various bias-conflicting ratios.
>
> > **Q4**: Missing recent studies such as [5, 6] in the related works section.
>
> **A4**: Thank you for letting us know. We will make sure to incorporate these recent studies in the revised version.
>
> [1] Just train twice: Improving group robustness without training group information, ICML 2021
>
> [2] Learning debiased representation via disentangled feature augmentation, NeurIPS 2021
>
> [3] Mitigating dataset bias by using per-sample gradient, ICLR 2023
>
> [4] Avoiding spurious correlations via logit correction, ICLR 2023
>
> [5] Surgical fine-tuning improves adaptation to distribution shifts, ICLR 2023
>
> [6] Fighting Fire with Fire: Contrastive Debiasing without Bias-free Data via Generative Bias-transformation, ICML 2023

---

> > ### Comment · Reviewer_qVjP · 2024-08-10
> >
> > Thank you for your detailed responses. Many of my concerns have been clarified.
> >
> > However, the concern regarding color augmentation still remains. As you mentioned, applying the same color augmentation to the baselines may ensure a fair comparison within that specific setting. However, this does not necessarily equate to an accurate evaluation of each method's debiasing effects. The vast majority of debiasing studies have generally omitted color augmentation because its inclusion can obscure an accurate assessment of the methods' effectiveness. It is well-known that color bias in colored MNIST and corruption bias in CIFAR-10C can be partially mitigated through color augmentation. Therefore, to ensure the reliability of future research on spurious correlations, I believe it is important to avoid using color augmentation in experimental settings.
> >
> > Given these concerns, I strongly recommend including main results without color augmentation (a comparison between the proposed method and state-of-the-art methods would suffice, considering the limited time available) as well as ablation studies specifically addressing the impact of color augmentation. If this concern is addressed, I will raise my score. However, if not, I cannot overlook the contribution of color augmentation to the main results, and therefore, I cannot agree to accept this paper.

---

> > > ### Author Response · Authors · 2024-08-12
> > >
> > > We appreciate your comments.
> > >
> > > As you suggested, we conducted experiments on C-MNIST, CIFAR-10C, and additionally on the real-world dataset BFFHQ, after removing color jitter. We chose PGD [1] as the comparison baseline, as it was recently proposed and showed the highest performance among the baselines. We report the average accuracies (%) and standard deviations over three trials. The results are shown in the table below.
> > >
> > > |            | C-MNIST (0.5%) | C-MNIST (1%) | C-MNIST (5%) | CIFAR-10C (0.5%) | CIFAR-10C (1%) | CIFAR-10C (5%) | BFFHQ (Unbiased) | BFFHQ (Conflict) |
> > > |:----------:|:--------------:|:------------:|:------------:|:----------------:|:--------------:|:--------------:|:----------------:|:----------------:|
> > > |     ERM    |  60.19 (0.96)  | 79.01 (0.56) | 95.23 (0.07) |   22.90 (0.76)   |  25.94 (0.69)  |  39.14 (0.44)  |   77.80 (0.61)   |   56.00 (0.35)   |
> > > |     PGD    |  96.83 (0.13)  | 98.14 (0.11) | 98.40 (0.07) |   30.01 (1.25)   |  41.55 (0.51)  |  52.17 (0.37)  |   84.17 (1.38)   |   70.20 (1.91)   |
> > > | DPR (Ours) |  97.45 (0.14)  | 98.28 (0.13) | 98.44 (0.21) |   31.91 (0.55)   |  43.31 (1.01)  |  52.92 (0.20)  |   87.13 (0.87)   |   76.53 (2.20)   |
> > >
> > > As you can see in the table above, even without color jitter, the proposed DPR outperforms ERM and PGD on all benchmarks (i.e., C-MNIST, CIFAR-10C, and BFFHQ). Additionally, we would like to mention that we followed the experimental settings of CNC and JTT respectively for CelebA and CivilComments-WILDS, which did not use any data augmentation. Therefore, color jitter was not used on them. Considering these points, we believe that DPR clearly demonstrates its effectiveness for mitigating spurious correlations.
> > >
> > > In addition, we would like to address the impact of color jitter in the experimental settings. As you can see from the results in the table above, the experimental results for CIFAR-10C shown in the previous answer, and Tables 1 and 2 in the paper, the presence or absence of color jitter does make a difference, but the difference is somewhat smaller than expected. We believe this may be due to the low intensity of color jitter used for data augmentation. For datasets other than CelebA and CivilComments-WILDS, we followed the experimental settings of Ahn et al. [1], which include various data augmentations including color jitter. The code for the color jitter used in the experiments is as follows, and you can find this in the code submitted with the paper as supplementary material.
> > >
> > > torchvision.transforms.ColorJitter(hue=0.05, saturation = 0.05)
> > >
> > > As shown in the code above, only hue and saturation were set, both at 0.05, and this was used for all experiments on C-MNIST, MB-MNIST, BAR, and BFFHQ. We believe that because a fairly low value of 0.05 was used for color jitter, it likely did not significantly alter the color characteristics of the data images. Furthermore, we think that even without color jitter, other data augmentations such as random rotation and random resize crop were sufficient to increase the diversity of bias-conflicting samples, thus achieving high debiasing performance.
> > >
> > > [1] Mitigating dataset bias by using per-sample gradient, ICLR 2023

---

> > > > ### Comment · Reviewer_qVjP · 2024-08-13
> > > >
> > > > I sincerely appreciate the additional experiments conducted in a short time, which have addressed many of my concerns. As a result, I will raise my score.
> > > >
> > > > Also, I strongly recommend that the issues discussed during the discussion period, particularly those regarding color augmentation, be incorporated into the revision to guide future research on spurious correlations in a more rigorous direction

---

> > > > > ### Author Response · Authors · 2024-08-14
> > > > >
> > > > > Thank you for taking the time to read our responses and provide numerous comments on them. As you mentioned, we intend to incorporate the issues discussed during the discussion period into the revised version. We are very grateful for your decision to raise the score.

---

### Author Rebuttal · Authors · 2024-08-07

We are very grateful to all the reviewers for their valuable comments.

The additional figures showing the experimental results of group identification for BFFHQ are in the PDF file.

---

### Decision · Program_Chairs · 2024-09-25

**Decision:**

Accept (poster)

**Comment:**

The paper proposes Disagreement Probability based Resampling for debiasing to mitigate the effects of spurious correlations without requiring explicit bias labels. The procedure uses a biased model to identify bias-conflicting samples and upsamples them.

The reviewers agreed the paper addresses a significant problem and appreciated the simplicity and practicality of the proposed solution. The procedure comes with both theoretical guarantees and extensive empirical results. The reviewers' concerns were sufficiently addressed in the author response and all were in favor of acceptance.